# Drug Repurposing Approaches towards Defeating Multidrug-Resistant Gram-Negative Pathogens: Novel Polymyxin/Non-Antibiotic Combinations

**DOI:** 10.3390/pathogens11121420

**Published:** 2022-11-25

**Authors:** Augustine Koh Jing Jie, Maytham Hussein, Gauri G. Rao, Jian Li, Tony Velkov

**Affiliations:** 1Department of Biochemistry and Pharmacology, School of Biomedical Sciences, Faculty of Medicine, Dentistry and Health Sciences, The University of Melbourne, Parkville, VIC 3010, Australia; 2Monash Biomedicine Discovery Institute, Department of Microbiology, Monash University, Clayton, VIC 3800, Australia; 3Division of Pharmacotherapy and Experimental Therapeutics, Eshelman School of Pharmacy, University of North Carolina, Chapel Hill, NC 27599, USA

**Keywords:** antimicrobial resistance, polymyxins, drug repurposing, non-antibiotic agents

## Abstract

Multidrug-resistant (MDR) Gram-negative pathogens remain an unmet public health threat. In recent times, increased rates of resistance have been reported not only to commonly used antibiotics, but also to the last-resort antibiotics, such as polymyxins. More worryingly, despite the current trends in resistance, there is a lack of new antibiotics in the drug-discovery pipeline. Hence, it is imperative that new strategies are developed to preserve the clinical efficacy of the current antibiotics, particularly the last-line agents. Combining conventional antibiotics such as polymyxins with non-antibiotics (or adjuvants), has emerged as a novel and effective strategy against otherwise untreatable MDR pathogens. This review explores the available literature detailing the latest polymyxin/non-antibiotic combinations, their mechanisms of action, and potential avenues to advance their clinical application.

## 1. Introduction

Before the modern era of antibiotics, bacterial infections such as pneumonia, tuberculosis, and gastrointestinal infections were responsible for high fatality rates worldwide [1]. Following the introduction of penicillin, the world has witnessed the clinical approval of a plethora of antibiotic classes that have extensively decreased mortality rates, and improved human quality of life [2,3]. Regrettably, the misuse and excessive administration of antibiotics in healthcare and agriculture have led to the selection and proliferation of multi-drug resistant (MDR) strains, including MDR Gram-negative pathogens (e.g., *Acinetobacter baumannii*, *Pseudomonas aeruginosa* and *K. pneumoniae*) [1,3,4]. These MDR Gram-negative bacteria have been recently ranked by the World Health Organization as critical priority pathogens, owing to their prevalence behind most hospital-acquired infections (accounting for 70% since 2015) and the shortage of relevant effective antibiotics to treat them [5]. Presently in the United States, MDR infections have led to ~35,000 fatalities annually, and it is projected that by 2050, the annual mortality rate of antibiotic resistance (10 million) would supersede even that of cancer (8.2 million) [6,7]. Antibiotic resistance is achieved via genetic mutations and by horizontal gene transfer of resistance genes between bacteria, augmenting the pathogen’s viability in environmental niches previously perilous for survival [4,8]. Resistant pathogens utilize four mechanisms to counter antibiotic action: (i) Active expulsion of antibiotics via the overexpression of membrane efflux pumps, (ii) mutations in membrane porin structure or decreased porin expression to decrease antibiotic uptake, (iii) expression of antibiotic-degradative or inactivating enzymes, and (iv) structural modification of drug targets impeding antibiotic binding [9].

Exacerbating this troublesome situation is the decline of new antibiotic approvals over the past 30 years, largely because drugs used for treating chronic diseases (i.e., cancer and metabolic diseases) were deemed to have far better profitability compared to antibiotics; the latter exhibit therapeutic activity of short duration, and rapidly become obsolete when resistant pathogens become prevalent in the community over time [1,3,9]. Without effective antibiotics, invasive and immunosuppressive treatments would become prohibitive, and healthcare systems would find themselves extensively overwhelmed by the spread of MDR bacteria [1,4]. Hence, while prudent stewardship of antibiotics may mitigate the spread of antibiotic resistance, it remains paramount that we expand our armamentarium of alternative treatment options to compensate for the dwindling efficacies of various last-resort antibiotics, such as polymyxins and carbapenems, which are used to treat persistent Gram-negative infections [10]. Drug repurposing (also known as drug repositioning) has been receiving heightened attention as an effective strategy to reassess de-risked FDA-approved drugs and/or investigational drugs for new therapeutic purposes [11]. This serendipitous approach offers lower overall development costs, shorter development timelines, and extremely low safety risks, compared to the *de novo* discovery of new agents (e.g., antibiotics) [12,13]. For decades, combining conventional antibiotics with non-antibiotics has been successfully used to treat infections caused by problematic Gram-negative and Gram-positive pathogens, tuberculosis, malaria, and viruses [14].

In this review, we shed light on the progress of the last-resort antibiotics polymyxins and non-antibiotic combinations, in combating the MDR Gram-negative pathogens based on the available literature, highlighting these non-antibiotics as prospective adjuvants. Further perspectives on translating in vitro synergistic polymyxin/non-antibiotic combinations to the same effect *in vivo*, to yield favorable clinical outcomes, will be discussed as well.

## 2. Combination Strategies to Overcome Antibiotic Resistance

### 2.1. Antibiotic-Antibiotic Combination Therapy

Antibiotic–antibiotic combinations have been explored and used as a popular treatment regime, as necessitated by the severity of the blight of antibiotic resistance. The reason behind using combinations is that the general probability of developing resistance against multiple antibiotics simultaneously, is surmised to be minimal; therefore, this enables rapid eradication of resistant pathogens before resistance to either or both drugs used in combination can arise, as the resulting synergistic or additive antibacterial effect of the combined antibiotics exceeds that of either drug alone [15,16]. With different metabolic pathways and/or drug targets inhibited concurrently, it increases the spectrum of coverage and potentially reduces the dosing regimens required for each antibiotic involved [17,18]. One notable antibiotic combination that was approved by the FDA to treat MDR-tuberculosis infections was the bedaquiline, pretomanid and linezolid cocktail [19,20]. This combination exerts synergistic bactericidal activity via the simultaneous disruption of oxidative phosphorylation (bedaquiline), protein synthesis inhibition (linezolid), mycolic acid biogenesis inhibition and/or release of reactive nitric oxide (pretomanid) [21,22,23]. However, certain antibiotic combinations can have an unintended antagonistic effect; one antibiotic could indirectly induce resistance against a second antibiotic, and potentially result in super-infections due to the survival of resistant bacteria [24]. Furthermore, resistance can evolve against both antibiotics concurrently, owing to horizontal gene transfer events [25]. To illustrate, Vestergaard et. al., observed that the combination of ciprofloxacin and ceftazidime augmented resistance in *Pseudomonas aeruginosa* compared to either antibiotic used alone. The combination resulted in overexpression of the mexAB–oprM operon; thus, increasing the production of efflux pumps, resulting in increased efflux of drugs out of the bacterial cell, and lowering the intracellular concentration [26]. Aside from substantially increased monitoring, drug preparation and administration costs associated with antibiotic co-administration, toxic side effects can also be exacerbated following exposure to multiple antibiotics. The latter is exemplified by the combined use of aminoglycosides and cephalosporins, which augment nephrotoxicity [27,28]. Therefore, the development of antibiotic alternatives or the combined use of antibiotics with non-antibiotic drugs that reverse bacterial resistance, is of the utmost importance [15].

### 2.2. Drug Repurposing

Drug repurposing is a cost-efficient strategy involving the investigation of existing approved drugs and/or investigational drugs for new therapeutic purposes, such as antimicrobial therapy [11,29]. Through this strategy, failures in drug discovery are mitigated as, for example, certain drugs withdrawn because of safety concerns could be salvaged for treating alternate diseases. One notable example is the teratogen thalidomide, which was successfully repurposed as a first-line antineoplastic for multiple myeloma because its antiangiogenic activity was confirmed to be therapeutically advantageous by destroying blood vessels nourishing malignant tumors—despite being responsible for arrested fetal limb development [30]. Another drug that is successfully repurposed is cyclosporin A, an immunosuppressant for treating organ transplantation and autoimmune diseases, which displays antibiofilm activity against *Mycobacterium tuberculosis*, enabling its clinical application as an anti-tuberculosis therapeutic [31,32]. Moreover, as the drug candidates selected for repurposing have their pharmacokinetic/pharmacodynamic profiles established from prior discovery and preclinical stages, they have already passed most of the toxicological and pharmacokinetic safety hurdles, allowing drug developers to bypass approximately 40% of the overall cost of commercializing their candidates [33,34].

Hence, drug repurposing is an attractive option for filling the current antibiotic discovery void [35]. However, repurposing selected drug candidates as antibacterial agents does bring about its own challenges. To illustrate this point, we would like to emphasize that antibiotics are typically administered at considerably higher dosages than non-antibiotic drugs; therefore, efficacy is determined at doses exceeding the prior effective concentrations for non-infectious diseases that the drug candidate was originally employed for therapeutically, thus posing toxicity risks. In addition, plasma protein binding often results in insufficient free drug concentrations at the infection site, impairing antibacterial activity [12]. A means to circumvent these drawbacks is combination therapy, of the non-antibiotic drugs of interest with existing antimicrobials, an approach that would prove advantageous due to the dose-sparing impact of synergistic combinations and/or the reversal of bacterial resistance [36].

### 2.3. Antibiotic and Non-Antibiotic/Adjuvant Combination Therapy

Certain co-administered non-antibiotic drugs despite being conventionally used to treat non-infectious diseases and possessing little to no in vitro antibacterial activity, remarkably are able to re-sensitize resistant pathogens to antibiotics [37,38]. It is purported that such non-antibiotics, designated adjuvants, potentiate antibiotic activity by interfering with the modes of resistance involving intrinsic mechanisms such as enzyme inhibition, obstructing efflux pumps, increasing membrane permeability, interfering with virulence signaling pathways, and biofilm formation (Figure 1) [39,40]. For these reasons non-antibiotic compounds could be repurposed for use in antibiotic/non-antibiotic combination therapies, to compensate for the severe attrition experienced in the antibiotic pipeline [13,33].

Multiple compounds with adjuvant properties were since discovered and semi-synthetically developed, functioning as efflux pump inhibitors, membrane-permeabilizing peptides and antibiofilm agents (with brief examples outlined in Table 1). Nonetheless, the antibiotic/adjuvant combinations that have successfully entered clinical use, to date, are the β-lactam/β-lactamase inhibitor combinations and the siderophore-cephalosporin conjugate cefiderocol (Table 1). β-lactams are popularly prescribed antibiotics that exert bactericidal action by inhibiting peptidoglycan chain cross-linking [42]. Although the β-lactamase inhibitors themselves do not kill target bacteria, they augment β-lactam efficacy by competitively inhibiting β-lactamases, shielding the β-lactam from degradation [43]. Similarly, the catechol scaffold of cefiderocol is devoid of intrinsic antibacterial activity but increases cephalosporin uptake via the bacterial membrane iron transport pathways [44]. Hence, adjuvant strategies display promising potential in circumventing the constant need for new antibiotics to overcome resistance, by acting as decoys that retard the activity of resistance-conferring enzymes.

## 3. The Time-Honored ‘Magic Bullet’ Polymyxin Lipopeptide Antibiotics Are Rapidly Losing Their Caliber

Polymyxins were discovered in the 1940s and introduced in the clinic as effective antibacterial agents against Gram-negative bacteria. However, their clinical use was reserved in the 1960s owing to their nephrotoxicity and neurotoxicity, and also better tolerated antibiotics from different classes were introduced [56]. In recent times, the use of polymyxins has experienced a resurgence as a treatment of last resort against difficult-to-treat MDR Gram-negative pathogens; however, alarming reports from clinics showed that there is a sharp upsurge in the polymyxin resistance particularly following polymyxin monotherapy [57,58,59]. This would warrant a necessity to develop an innovative strategy (e.g., combining polymyxins with nonantibiotics) to preserve the clinical effectiveness of polymyxin and halt the development of polymyxin resistance [60,61]. In terms of their chemical constitution, polymyxins are non-ribosomal cyclic lipopeptides extracted from the Gram-positive soil bacterium *Paenibacillus polymyxa*. Fundamentally, polymyxins are comprised of a polycationic ring with a short peptide extension to which a lipophilic chain is attached [57]. Of the five major polymyxin isoforms (A–E), only two (polymyxin B and polymyxin E aka. colistin) were clinically utilized in the 1950s [62]. Polymyxin B and colistin share nearly identical structures, only differing by an amino acid at position six in the peptide ring (Figure 2A), and as such share cross-resistance [57,62]. Polymyxins bind the lipid A component of LPS via electrostatic attractions, displacing the Ca^2+^ and Mg^2+^ ions required to stabilize the LPS leaflet. Thus, the polymyxin is anchored in the outer membrane via its lipophilic chain, causing membrane disruption. Subsequent polymyxin-mediated fusion of the inner leaflets of the outer membrane and the outer leaflet of the cytoplasmic membrane, results in osmotic lysis (Figure 2B) [62,63,64]. Other proposed secondary mechanisms leading to bacterial death include polymyxin inhibition of NADH oxidases involved in cell respiration, and the induction of reactive oxygen species (ROS) production [65]. However, owing to their membrane-disrupting activities, aside from nephrotoxicity polymyxins also cause neurotoxicity in the host, so their use is unduly restricted. Furthermore, the lack of properly established optimized regimens resulted in suboptimal drug exposure selecting for polymyxin-resistant strains [66].

The Gram-negative bacterial outer membrane serves as the primary means of resistance against multiple antibiotic classes; any alteration in the outer membrane, such as increased hydrophobicity, mutations in porins, overexpression of efflux pumps, and other factors, significantly impedes antibiotic penetration [67]. The outer membrane is comprised of a phospholipid-rich inner leaflet, akin to the cytoplasmic inner membrane, and the LPS rich outer leaflet which faces the extracellular environment [68]. The LPS contains three key domains, namely the *O*-antigen, 2-keto-3-deoxyoctonoic acid (inner core), and lipid A, which possesses anionic phosphate moieties accentuating the outer membrane’s selective permeability [69].

The primary resistance mechanisms employed by polymyxin-resistant strains include the synthesis of modified lipid A with anionic charges neutralized, preventing LPS binding by polymyxins. Succeeding events such as the cessation of LPS production, as well as the upregulation of genes expressing phospholipids and other OM components, augment polymyxin resistance [63,66]. Furthermore, a plasmid-borne mobile colistin resistance gene (*mcr-1*), a gene that was first reported in China, which upon expression culminates in the modification of lipid A with phosphoethanolamine, impairing polymyxin targeting [70]. The *mcr-1* plasmid is readily transferrable via conjugation across Gram-negative species, which results in the global dissemination of *mcr-1*, therefore, substantially threatening polymyxin efficacy and raising public health concerns globally [71,72]. Hence, various research groups and our own have investigated combinations of several FDA-approved non-antibiotic drugs and natural products with polymyxins, to evaluate possible synergy against MDR/polymyxin-resistant Gram-negative pathogens and the repurposing potential of these non-antibiotics as polymyxin adjuvants. Instances of non-antibiotic drugs tested in combination with polymyxins are listed in Table 2.

## 4. Polymyxin/Non-Antibiotic Combinations

### 4.1. Antineoplastic Drugs

Our group firstly reported the antibacterial synergetic activity of three mainstream selective estrogen receptor modulators (SERMs), tamoxifen, raloxifene, and toremifene in combination with polymyxin B [73,74]. While conventionally used for breast cancer treatment, some SERMs were reported to exhibit direct antimicrobial properties via interference with cell wall synthesis (e.g., inhibition of wall teichoic acid synthesis in *E. faecium* and *S. aureus* by clomiphene) and membrane perforation (e.g., tamoxifen) [96,97]. Based on time-kill assays, polymyxin-resistant *P. aeruginosa*, *Klebsiella pneumoniae*, and *Acinetobacter baumannii* strains were considerably re-sensitized to polymyxin B, in the presence of each SERM (≥2.0-log_10_ decrease in bacterial viability relative to either monotherapy at 24 h) [74]. Metabolomics analysis of the tamoxifen-polymyxin B combination against an MDR polymyxin-resistant cystic fibrosis *P. aeruginosa* (FADDI-PA006) strain revealed extensive perturbations in the fatty acid and glycerophospholipid synthetic pathways involved in membrane biogenesis; thus implying the combination exerts synergy via concerted damaging effect against the outer membrane [73].

Mitotane, an antineoplastic drug used for the treatment of adrenal cancers [98], displayed synergy with polymyxin B against polymyxin-resistant MDR *A. baumannii*, *P. aeruginosa*, and *K. pneumoniae* in vitro. It also prevented the development of polymyxin-resistance (i.e., regrowth) in the polymyxin-susceptible isolates, compared to monotherapy from time-kill assays [75]. The in vitro synergy was further confirmed in a mouse burn wound model, in which the combination treatment decreased wound infection by a polymyxin-resistant *A. baumannii* isolate, compared to either polymyxin B or mitotane monotherapy. Scanning electron microscopy and transmission electron microscopy (SEM/TEM) imaging of *A. baumannii* isolates ATCC™ 17978 and FADDI-AB225 treated with the combination, revealed the formation of aberrant cell clusters suggesting the combination disrupts binary fission. While outer membrane damage was observed in the polymyxin-susceptible ATCC™ 17978 following polymyxin monotherapy, the polymyxin-resistant FADDI-AB225 strain displayed membrane blebbing; thus, it was inferred that polymyxin B perforates the outer membrane sufficiently for the hydrophobic mitotane to diffuse across the outer membrane and exerts antibacterial activity against intracellular targets. Untargeted metabolomics analysis revealed significant disruptions in the TCA cycle and nucleotide metabolism in the metabolome of four *A. baumannii* isolates treated with the combination [76].

### 4.2. Antipsychotic and Antidepressant Agents

Phenothiazines are heterocyclic compounds clinically utilized as antipsychotics, that function as dopamine antagonists [99,100]. Over time, the antimicrobial characteristics of the phenothiazines were serendipitously discovered, such as chlorpromazine which was observed to exert anti-mycobacterial activity and thioridazine which potentiates first-line anti-tuberculosis antibiotics enabling rapid clearance of MDR/XDR (extensively drug-resistant) tuberculosis infections [99,101,102,103]. The proposed antibacterial mechanisms exerted by phenothiazines include augmenting complete phagocytosis of bacteria by macrophages, inducing abnormalities in binary fission, and inhibiting antibiotic efflux pumps [99,100,101]. The three phenothiazines, prochlorperazine, chlorpromazine, and thiethylperazine were investigated by our group [78]. Among the three, prochlorperazine showed superior synergy with polymyxin B, as evident from the rapid and extensive bactericidal activity (4.0–6.0-log_10_ CFU/mL decrease in bacterial inoculum at 4 and 8 h, relative to the untreated control) observed against all tested strains in the time-kill assays. SEM/TEM imaging revealed extensive membrane blebbing, cell shriveling, perforation, and vesicle formation following exposure of the polymyxin-resistant *P. aeruginosa* strain FADDI-PA070 to the polymyxin B/prochlorperazine combination, reflecting augmented damage to the outer membrane. This was further verified by metabolomics, which reveal considerable perturbations in glycerophospholipid, fatty acid, and LPS biosynthesis, following exposure to the combination. Additional pathways perturbed include acetyl-CoA, arginine, and proline metabolism, suggesting the combination also impairs bacterial respiration, growth, and biofilm formation [78].

Selective serotonin reuptake inhibitors (SSRI) function by blocking presynaptic axon terminal serotonin transporters, and are largely prescribed for major depressive and anxiety disorders [104]. Interestingly, various research groups have reported antimicrobial activity from SSRIs, such as femoxetine and paroxetine, that purportedly exert direct antibacterial activity via efflux pump inhibition [105]. One SSRI in particular, sertraline, was reported by Ayaz et. al., to exert concentration-dependent reversal of resistance towards ciprofloxacin, levofloxacin, norfloxacin, gentamicin, and moxifloxacin in the clinical isolates of *S. aureus*, *E. coli*, and *P. aeruginosa* [104]. Sertraline was observed to readily synergize with polymyxin B at lower concentrations in time-kill assays (>3.0-log_10_ CFU/mL decrease in bacterial viability) against polymyxin-resistant *P. aeruginosa*, *A. baumannii*, and *K. pneumoniae* strains [82]. Further SEM/TEM and metabolomics studies indicated the combination synergistic killing activity primarily involves the inhibition of amino-sugar, sugar-nucleotide metabolism, glycerophospholipid, and fatty acid metabolism causing disruption of the outer membrane integrity, cell wall synthesis, and respiration [82].

### 4.3. Antifungal Drugs

Caspofungin is an antifungal drug that operates by inhibiting the enzyme (1→3)-β-D-glucan synthase, destabilizing fungal cell wall integrity. Caspofungin serves as a therapeutic agent against *Aspergillus* and *Candida* fungal infections [106]. Intriguingly, caspofungin was observed to exert antibacterial activity by inhibiting biofilms [80]. Further untargeted metabolomics analysis by Hussein et. al. [79] revealed the synergistic combination of polymyxin B/caspofungin against *K. pneumoniae* acts by inhibiting multiple interconnected metabolic pathways, including bacterial envelope biosynthesis, the phosphotransferase system (involved in biofilm formation), ATP-binding cassette (ABC) transporter production. Another antifungal, miconazole, possesses broad spectrum fungicidal activity by penetrating the chitin fungal cell wall and permeabilizing the cell membrane to external noxious substances [107]. This membrane-disrupting ability of miconazole allows it to interfere with bacterial lipid membranes and accounts for its direct antibacterial activity [108]. The combination of polymyxin B/miconazole was reported to exert synergistic bacterial killing, purportedly through permeabilization of the outer membrane by polymyxin B, which in turn allows miconazole to access the periplasmic space and disrupt the inner cytoplasmic membrane, causing bacterial lysis [81].

### 4.4. Antiparasitic Drugs

Closantel exerts an anti-parasitic mechanisms of action via the disruption of oxidative phosphorylation and the inhibition of chitinase activity [109]. As monotherapy, closantel had no antibacterial effect (minimal inhibitory concentration, MIC ≥ 16 mg/L); however, in combination with polymyxin B, synergy and inhibition of polymyxin resistance against *A. baumannii* were achieved at therapeutic concentrations [88]. In the presence of closantel, the polymyxin-resistant *A. baumannii* strains (polymyxin MIC ≥ 4 mg/L) were considerably re-sensitized to polymyxin B concentrations at 2 mg/L and below, which coincides with the polymyxin MIC susceptibility breakpoints (≤2 mg/L) [110]. In addition, Domalaon et. al., investigated the combination of colistin/closantel, and with other related anthelmintic salicylanilides oxyclozanide and rafoxanide. The combinations displayed synergy against MDR *P. aeruginosa*, *A. baumannii*, *K. pneumoniae*, *E. coli*, and *E. cloacae*, and reversed resistance to colistin apparently via potentiating damage to the outer membrane [91].

### 4.5. Natural Products

Cannabinoids are secondary metabolites isolated from the plant *Cannabis sativa*, the same plant from which the illicit drug marijuana is derived [111]. Several cannabinoids are behind the psychotropic effects of marijuana; these are synthesized via the alkylation of olivetolic acid, producing cannabigerolic acid, a precursor to most cannabinoids [112]. Of all the cannabinoids discovered, cannabidiol is the main non-psychoactive ingredient of *Cannabis sativa* [113]. Reports of antibacterial activity exerted by cannabidiol date back to the 1950s, with an investigation conducted by Van Klingeren and Ten Ham [114], noting MICs of cannabidiol within the ranges of 1–5 μg/mL for Gram-positive *Staphylococci* and *Streptococci*. Additional investigations carried out by Blaskovich et. al. [113] revealed consistent MICs (1–4 μg/mL) of cannabidiol against many Gram-positive MDR *Staphylococci* (methicillin-resistant *S. aureus*, vancomycin-intermediate *S. aureus*, vancomycin-resistant *S. aureus*), vancomycin-resistant *Enterococci* (*E. faecalis*, *E. faecium*), *Clostridioides* (*C. difficile*), and *Streptococci* (*S. pyogenes*, *S. pneumoniae*) strains. Radiolabeled macromolecular synthesis assays in *S. aureus* RN42200 and bacterial cytological profiling, indicated cannabidiol exerts antibacterial activity via membrane permeabilization, as evident from the marked perturbations in lipid synthesis at sub-MIC levels.

However, cannabidiol was inactive against Gram-negative species tested, with the exceptions of *Neisseria* and *Legionella* isolates. Further investigations involving exposure of efflux pump *E. coli* and *P. aeruginosa* mutants revealed the inactivity of cannabidiol nonetheless, ruling out efflux pumps as a possible explanation for the cannabinoid’s general ineffectiveness against Gram-negative bacteria. This was also confirmed by Abichabki et. al., who noted the inactivity of cannabidiol against *K. pneumoniae*, *A. baumannii*, *P. aeruginosa*, and *E. cloacae* even, in the presence of efflux pump inhibitors such as curcumin [84]. However, cannabidiol was active against an *E. coli* lpxC cell membrane mutant, indicating that outer membrane LPS deficiency enables penetration of cannabidiol. This conclusion was further supported by the increased susceptibility of lipid A-deficient *A. baumannii* (ATCC™ 19606R) to cannabidiol (MIC >128 to 0.25 μg/mL), which in turn has an polymyxin MIC (>128 μg/mL) relative to its parent strain (0.25 μg/mL) [115]. Our group investigated polymyxin B and cannabidiol as another possible antibiotic/adjuvant combination for treating polymyxin-resistant Gram-negative infections [83]. Broth microdilution tests involving polymyxin B combined with cannabidiol (fixed at 256 μg/mL) against 13 different Gram-negative pathogens (52 strains in total), yielded observable antibacterial activity of the combination against 47 of the 52 strains. Subsequent checkerboard assays combining polymyxin B/cannabidiol verified the synergistic activity, as evident from the low polymyxin B concentrations (≤2 µg/mL, within the EUCAST “susceptible” breakpoint) enabling cannabidiol to exert antibacterial activity at a minimal effective antibiotic concentration (MEAC) ≤4 µg/mL. Moreover, the polymyxin B-cannabidiol combination substantially decreased the viability of four *K. pneumoniae* strains (>2.0-log_10_ CFU/mL decline) relative to polymyxin B monotherapy at several time points, verifying the combination’s synergistic bactericidal activity. Similarly, checkerboard and time-kill assays confirmed synergy against *A. baumannii*, *P. aeruginosa*, and *K. pneumoniae*, both polymyxin-resistant and polymyxin-susceptible isolates alike. Moreover, it was inferred from metabolomics analysis of *A. baumannii* ATCC™ 19606 that the polymyxin B/cannabidiol combination substantially disrupts cell envelope biogenesis through a time-dependent perturbation of amino–sugar and nucleotide–sugar metabolism, and the pentose phosphate pathway. Consequently, peptidoglycan and LPS synthesis are disrupted owing to decreased levels of core membrane lipid components, phosphatidylethanolamine, phosphatidic acid, *sn*-glycero-3-phosphoethanolanime, and *sn*-glycerol 3-phosphate.

Curcumin, a phenolic compound isolated from the plant *Curcuma longa*, has been traditionally used for gastrointestinal treatments owing to its antioxidant and anti-inflammatory activities [116]. Intriguingly, this natural product was also reported to exhibit antimicrobial activity; various reported antibacterial mechanisms exerted by curcumin include membrane damage, efflux inhibition, biofilm disruption, and growth inhibition [117,118]. When combined with polymyxin B, considerable synergy was observed against polymyxin-resistant strains (as well as expanding the polymyxin activity spectrum coverage to Gram-positive bacteria) at reduced polymyxin MICs; the mechanisms purported to account for synergy involve outer membrane permeabilization by curcumin, allowing access to the inner cytoplasmic membrane by polymyxin B [85]. Furthermore, colistin in combination with curcumin was observed to reverse polymyxin-resistance in *A. baumannii* through outer membrane permeabilization by colistin, facilitating the entry of curcumin which triggers increased ROS production, severely impacting bacterial viability [95].

Tetrandrine is a bis-benzylisoquinoline alkaloid isolated from the herb *Stephania tetrandra* [119]. Although tetrandrine functions pharmacologically as a calcium channel blocker, it was also found to possess anti-inflammatory and antineoplastic properties through the inhibition of pro-inflammatory cytokine production and the induction of apoptosis in tumor cells, respectively [120]. Interestingly, tetrandrine exhibits antibacterial properties as observed from its ability to inhibit antibiotic efflux from MRSA and *Mycobacterium tuberculosis* [121,122]. Hence, when tested in combination with colistin against *mcr-1*-positive colistin-resistant *Salmonella*, tetrandrine was noted to enhance colistin activity by enhancing the permeability of the outer membrane through efflux inhibition sufficiently for colistin to act, as well as by disrupting oxidative phosphorylation (ATP production) and down-regulating *mcr-1* expression [93].

### 4.6. Other Non-Antibiotic Drugs

The cystic fibrosis transmembrane conductance regulator (CFTR) potentiator ivacaftor, commonly used to treat cystic fibrosis patients, exerts apparent antimicrobial activity via weak inhibition of DNA gyrase and topoisomerase IV enzymes, owing to certain structural similarities with quinolones. When combined with polymyxin B against *P. aeruginosa*, augmented damage against the outer membrane was observed via nitrocefin assay, SEM and TEM [87]. This was further substantiated by metabolomics analysis which revealed extensive phospholipid and LPS biosynthesis perturbations, following exposure to the polymyxin/ivacaftor combination [86]. Auranofin, an anti-rheumatic drug for treating arthritis, displays negligible anti-Gram-negative activity [123]; however, in combining with colistin, it results in extensive cell shrinkage and lysis as observed from SEM/TEM imaging, validating auranofin as a prospective adjuvant. In addition, the colistin/auranofin combination extensively counters colistin resistance in Gram-negative pathogens and substantially improves the survival rate in mouse peritoneal infection models [89].

Zidovudine is a nucleoside/nucleotide reverse transcriptase inhibitor that acts by incorporating into the elongating viral DNA strand, and slowing down the progression of HIV infection [124]. Intriguingly, antibacterial activity was also observed from zidovudine, likewise by acting as a DNA-chain terminator and arresting bacterial DNA replication [125]. When combined with polymyxin B, zidovudine extensively potentiated polymyxin activity against *K. pneumoniae*, ≥4.0-log_10_ CFU/mL in time-kill assay; ≥3.0-log_10_ CFU/mL in murine thigh infection model [77]. Similarly, the colistin/zidovudine combination was observed to reverse colistin resistance in time-kill assay (>3.0-log_10_ CFU/mL) with the absence of regrowth in the combination 24 h post-exposure, and displayed in vivo synergy against NDM-1 *K. pneumoniae* and *mcr-1*-positive *E. coli* in murine peritoneal infection models [90,126].

Melatonin, a hormone secreted by the pineal gland in the brain, functions to regulate circadian rhythms and blood pressure, thus is used medically to treat sleep disorders. Astonishingly, melatonin was reported to exert bacteriostatic (growth-inhibiting) activity, via the perturbation of key metabolic pathways and the expression of genes involved in cell division. Furthermore, the combination of colistin with melatonin substantially permeabilizes the outer membrane and inhibits efflux pump activity, hence re-sensitizing polymyxin-resistant Gram-negative pathogens to polymyxins [127]. Additional transcriptomics analysis involving *mcr-1*-positive *E. coli* reveals significant disruptions in the expression of genes involved in LPS modifications and efflux pump production, in the presence of the colistin–melatonin combination [94].

## 5. Perspectives and Future Directions

This review highlights the promising clinical approach of combining non-antibiotic FDA-approved drugs or natural products with polymyxins, for treating MDR Gram-negative infections. Considering the majority of the FDA-approved non-antibiotic drugs are already well-characterized pharmacokinetically and toxicologically, it would enable them to forgo clinical trials as information from prior safety studies can help guide design of future studies focused on evaluating their applicability as antibacterial adjuvants [12]. Consequently, our ability to translate these adjuvants in combination with polymyxins to the bedside, could potentially be accelerated due to reduced developmental costs by the leapfrogging of extensive prior preclinical development insights [34,128]. Moreover, as these non-antibiotic drugs do not exert direct inhibitory activity against specific bacterial molecular targets, evolutionary pressure on bacteria to evolve resistance would likely be eschewed [38,129]. Hence, non-antibiotic drugs represent a valuable novel reservoir of anti-infective adjuvants for combination therapy.

In general, the polymyxin/non-antibiotic combinations covered in this review function synergistically by augmenting penetrative damage against the outer membrane causing bacterial lysis. Alternatively, the outer membrane may be permeabilized sufficiently (either by the polymyxin or the adjuvant) for the combination to access the inner membrane, leading to either perforation of the inner membrane (and lysis) or diffusion across the membrane, substantially disrupting vital metabolic pathways (i.e., respiration, DNA replication, cell envelope maintenance) and/or repressing plasmid-mediated polymyxin resistance (Figure 3). In consequence, effective polymyxin concentrations for bacterial killing are decreased, reversing polymyxin resistance, and potentially reducing polymyxin dosing regimens required for the maximum clearance of pathogens. This would improve the polymyxin therapeutic index, since nephrotoxicity and neurotoxicity side effects are likely to be reduced [130].

In closing, this contemporary review provides detailed information about a potentially central hub for several novel combinations, which could be utilized as promising antibacterial therapies to bridge the antibiotic development gap. In the last decade, this innovative therapeutic approach has emerged as an area of great interest in the anti-infectives field and should be pursued to fruition vigorously.

## Figures and Tables

**Figure 1 pathogens-11-01420-f001:**
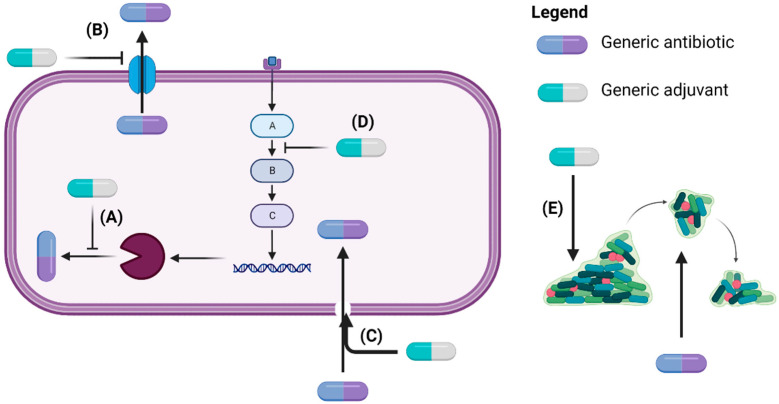
Adjuvants exert anti-resistance mechanisms by inhibiting enzymes involved in antibiotic inactivation (**A**), blocking efflux pumps (**B**), disrupting membrane integrity to enable intracellular antibiotic access (**C**), interfering with signaling pathways leading to antibiotic resistance (**D**), and dispersing bacteria in biofilms to expose them to antibiotics (**E**) (Created with BioRender.com) [39,41].

**Figure 2 pathogens-11-01420-f002:**
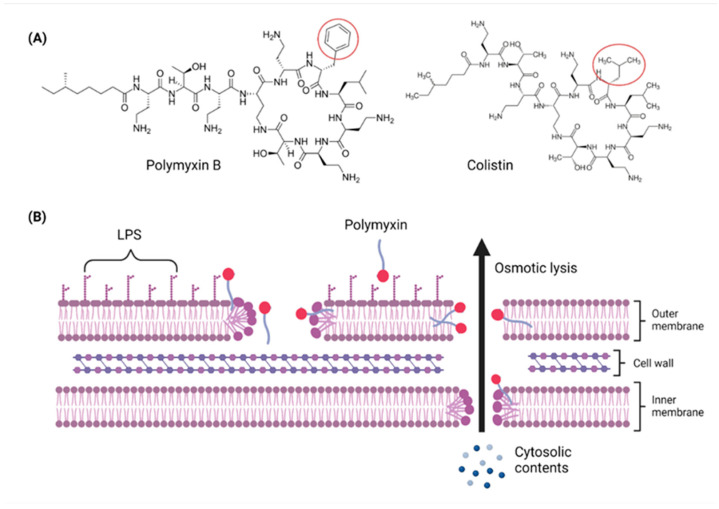
Chemical structures and pharmacological mechanisms of polymyxins. (**A**) Structures of polymyxin B and colistin with the amino acid differences between them circled in red. (**B**) The bactericidal action of polymyxins against Gram-negative bacteria via targeting LPS on the outer membrane and perforating the inner membrane (Created with Biorender.com).

**Figure 3 pathogens-11-01420-f003:**
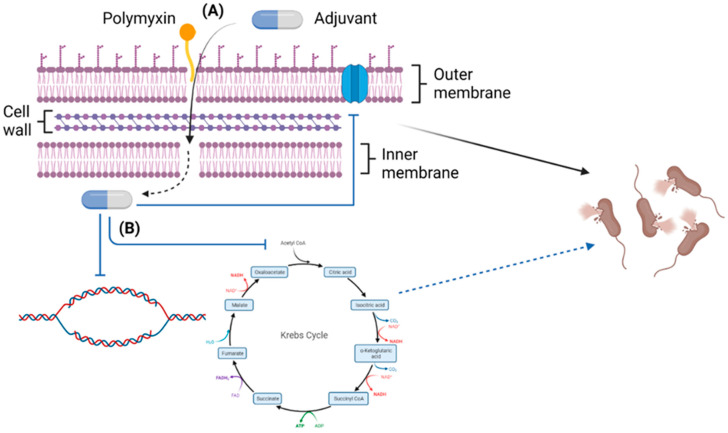
An overview of the general synergistic mechanisms of polymyxin/non-antibiotic combinations (Created with BioRender.com). (**A**) The combinations function via augmented membrane perforation, and (**B**) disruption of vital metabolic pathways, DNA replication, and gene expression contributing to resistance.

**Table 1 pathogens-11-01420-t001:** Examples of antibiotic/non-antibiotic combinations.

Mechanism of Action	Type of Adjuvant	Examples of Adjuvants	Antibiotic Potentiated	Commercial Combination	Reference
Antibiotic-inactivating enzyme inhibition	β-lactamase enzyme inhibitor	Avibactam	Ceftazidime	Avycaz^®^	Heo, 2021 [45] Drawz and Bonomo, 2010 [46] Bush and Bradford, 2016 [47]
		Vaborbactam (RPX7009)	Meropenem	Vabomere™	
		Tazobactam	Ceftolozane	Zerbaxa^®^	
			Piperacillin	Zosyn^®^	
		Clavulanic acid	Amoxicillin	Augmentin^®^	
			Ticarcillin	Timentin^®^	
		Sulbactam	Ampicillin	Unasyn^®^	
		Relebactam	Imipenem	Recarbrio™	
Membrane permeabilization	Peptides	Unacetylated tridecaptin	Vancomycin Rifamycin Erthryomycin	-	Cochrane et al., (2015) [48]
Increased antibiotic intracellular concentration	Efflux pump inhibitor	Tetracycline analogues	Tetracyclines	-	Van Bambeke et al., (2010) [49]
		Aminoglycoside analogues	Aminoglycosides		
		Polybasic peptide–levofloxacin conjugates Phenylalanine-arginine β-naphthylamide (PAβN)	Fluoroquinolones		Berry et al., (2019) [50] Lomovskaya et al., (2001) [51]
		Spectinamides	Clarithromycin Doxycycline Clindamycin		Bruhn et al., (2015) [52]
	Siderophores	Catechol	Ceftazidime	Cefiderocol (Fetroja^®^)	Zhanel et al., (2019) [44]
Damage to biofilms	Metal chelator	EDTA	Tetracycline, Ampicillin, Penicillin, Chloramphenicol, Ceftazidime, Gentamicin, Ciprofloxacin	-	Aboelenin et al., (2021) [53]
	Free radical species	Nitroxides	Ciprofloxacin Isothiazolone		Verderosa et al., (2019) [54] Verderosa et al., (2022) [55]

**Table 2 pathogens-11-01420-t002:** Summary of the polymyxin/non-antibiotic combinations investigated.

Polymyxin Class	Non-Antibiotic Adjuvant	Adjuvant Classification	Target Pathogen
Polymyxin B 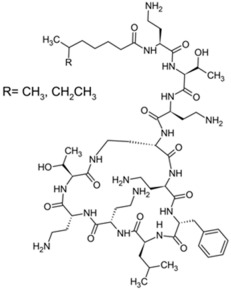	Tamoxifen [73,74] 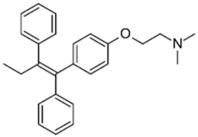	Antineoplastic (SERM *)	*K. pneumoniae*, *A. baumannii*, *P. aeruginosa*
	Raloxifene [74] 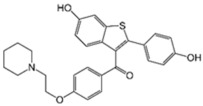		
	Toremifene [74] 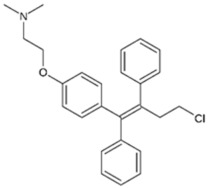		
	Mitotane [75,76] 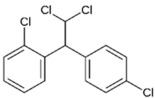	Antineoplastic	*K. pneumoniae*, *A. baumannii*, *P. aeruginosa*
	Zidovudine [77] 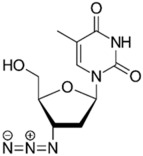	Antiretroviral	*K. pneumoniae,*
	Prochlorperazine [78] 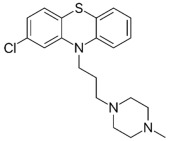	Antipsychotic	*K. pneumoniae*, *A. baumannii*, *P. aeruginosa*
	Thiethylperazine [78] 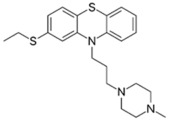		
	Chlorpromazine [78] 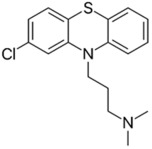		
	Caspofungin [79,80] 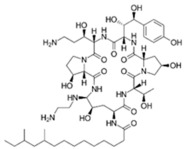	Antifungal	*K. pneumoniae*, *P. aeruginosa*
	Miconazole [81] 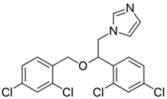		*E. coli*, *P. aeruginosa*
	Sertraline [82] 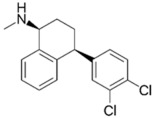	Antidepressant (SSRI **)	*K. pneumoniae*, *A. baumannii*, *P. aeruginosa*
	Cannabidiol [83,84] 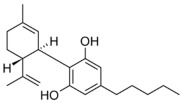	Natural Product (Cannabinoid)	*K. pneumoniae*, *A. baumannii*, *P. aeruginosa*, *N. gonorrhoeae*, *N. meningitidis*, *M. catarrhalis*
	Curcumin [85] 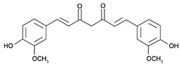	Natural Product	*A. baumannii*, *E. coli*, *P. aeruginosa*, *S. maltophilia*
	Ivacaftor [86,87] 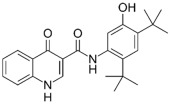	CFTR *** potentiator	*P. aeruginosa*
	Closantel [88] 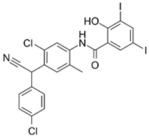	Anti-parasitic	*K. pneumoniae*, *A. baumanii*, *P. aeruginosa*, *E. coli*, *E. cloacae*
Colistin 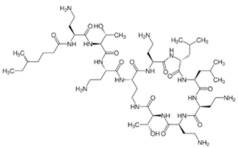	Auranofin [89] 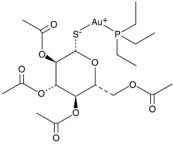	Anti-rheumatic	*K. pneumoniae*, *A. baumannii*, *P. aeruginosa*, *E. coli*
	Zidovudine [90] 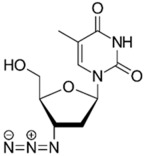	Antiretroviral	*K. pneumoniae*, *E. coli*, *E. cloacae*
	Rafoxanide [91] 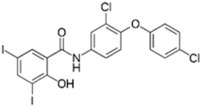	Anti-parasitic	*K. pneumoniae*, *A. baumanii*, *P. aeruginosa*, *E. coli*, *E. cloacae*
	Oxyclozanide [91,92] 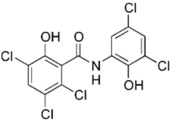		
	Tetrandrine [93] 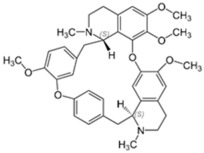	Natural product (bisbenzylisoquinoline alkaloid)	*Salmonella*
	Melatonin [94] 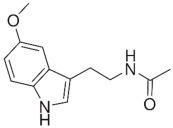	Endogenous hormone	*E. coli*
	Curcumin [95] 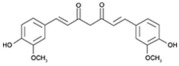	Natural Product	*A. baumannii*

* Selective Estrogen Receptor Modulator, ** Selective Serotonin Reuptake Inhibitor, *** Cystic Fibrosis Transmembrane Conductance Regulator.

## Data Availability

Not applicable.

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
