# Peer review of "Drug Repurposing Approaches towards Defeating Multidrug-Resistant Gram-Negative Pathogens: Novel Polymyxin/Non-Antibiotic Combinations"

_pathogens, 2022, doi:10.3390/pathogens11121420_

Round 1
Reviewer 1 Report
This article deals with a very interesting topic such as the drugs repurposing which is a cost-efficient strategy involving the investigation of existing approved drugs for new therapeutic purposes such as anti- microbial therapy. The fact to use compounds already on the market for other kind of diseases such as non-infectious diseases is a great advantage because these products have already been apprved by FDA so that we know their possible toxicity, their pharmacokinetics and pharmacodynamics characteristics, their mechanism of activity etc In this article the authors reviewed scrupulously and in detail some non-antibiotic compounds (ie antineoplastic drugs, antipsychotic and antidepressant agents, antiparasitic drugs, natural products and others) that could be associated with antibiotics increasing their activity and being able to re-sensitize the antimicrobial agents. The antibiotics considered by the authors as the "magic bullet" were the polymixins A and E (colistin). Even though these non-antibiotic products don't show amy antibacterial activity by themselves , however they make the antibiotics act against the bacteria, maybe inducing an osmotic lysis of the bacterial outer membrane and allowing them to penetrate inside the cells.
This article is interesting enough, well written, well organized and in a good English. The only concern should be the fact that the authors don't mention at all the in vivo activity of these combinations They result active in vitro showiing low MICs under the breakpoints of polymixins ( 2 μg/mL) but nothing is said about the activity in vivo. The authors should explain this issue.
Reviewer 2 Report
Please see attached file.

Reviewer 3 Report
Dear authors, Please find the attached file.

Round 2
Reviewer 3 Report
Dear authors
I have concerns regarding some comments in my first revision needed to be addressed such as the causes beyond the choice of the authors for some antibiotics and some mechanisms. Please, refer to the previous revision and address my comments.
